# A one-gate elevator mechanism for the human neutral amino acid transporter ASCT2

Alisa A. Garaeva[1], Albert Guskov [2], Dirk J. Slotboom [1,3] & Cristina Paulino [1,2]

The human Alanine Serine Cysteine Transporter 2 (ASCT2) is a neutral amino acid exchanger that belongs to the solute carrier family 1 (SLC1A). SLC1A structures have revealed an elevator-type mechanism, in which the substrate is translocated across the cell membrane by a large displacement of the transport domain, whereas a small movement of hairpin 2 (HP2) gates the extracellular access to the substrate-binding site. However, it has remained unclear how substrate binding and release is gated on the cytoplasmic side. Here, we present an inward-open structure of the human ASCT2, revealing a hitherto elusive SLC1A conformation. Strikingly, the same structural element (HP2) serves as a gate in the inward-facing as in the outward-facing state. The structures reveal that SLC1A transporters work as one-gate elevators. Unassigned densities near the gate and surrounding the scaffold domain, may represent potential allosteric binding sites, which could guide the design of lipidic-inhibitors for anticancer therapy.

[1] Membrane Enzymology, University of Groningen, Groningen Biomolecular Sciences and Biotechnology Institute, Groningen, The Netherlands. [2] Structural Biology, University of Groningen, Groningen Biomolecular Sciences and Biotechnology Institute, Groningen, The Netherlands. [3] University of Groningen, Zernike Institute for Advanced Materials, Groningen, The Netherlands. Correspondence and requests for materials should be addressed to D.J.S. (email: d.j.slotboom@rug.nl) or to C.P. (email: c.paulino@rug.nl)

The Alanine Serine Cysteine Transporter 2 (ASCT2) belongs to the solute carrier family 1 (SLC1A) of transporters, which in mammals consists of the excitatory amino acid transporters EAAT1–EAAT5 (SLC1A1-3, 6–7) and the neutral amino acid transporters ASCT1-ASCT2 (SLC1A4-5)[1]. Although EAATs and ASCTs share significant similarities in sequence (~40%) and structure[2], they are mechanistically different: EAATs typically concentrate the neurotransmitter ʟ-glutamate against its gradient[3]; whereas ASCTs catalyse the exchange of neutral amino acids[4].

Over the past years, ASCT2 has increasingly gained attention as a potential target in anticancer therapy, as it is upregulated and linked to poor survival in melanoma, lung, breast, prostate, pancreatic, thyroid and colon cancer[5–10]. ASCT2 plays a role in cancer cell growth by providing glutamine as an alternative carbon source for the tricarboxylic acid (TCA) cycle, for fatty-acid production, and by contributing to the activation of the mammalian target of rapamycin complex (mTORC1)[11]. ASCT2 also acts as a receptor for several retroviruses[12], and a recently determined structure[2] revealed unique extracellular antennae that likely form the required recognition platform[13].

Structures of the prokaryotic SLC1A transporters Glt_Ph and Glt_Tk and the human EAAT1 and ASCT2, in different conformational states, have provided insights into the transport cycle[14–19]. They all share a similar homotrimeric protein architecture, where each protomer consists of a scaffold and a transport domain. The scaffold domains provide the trimer interface and do not move during the transport cycle. By contrast, each transport domain located at the periphery binds the amino acid and coupling ions, and translocates the substrates across the membrane in an elevator-like manner. During this movement the substrate is transported through the lipid bilayer in a fully occluded state. Structures obtained in outward-open and outward-occluded states, have revealed how the substrate reaches the binding site from the extracellular side via a movement of hairpin 2 (HP2)[15,19]. However, it remains unknown how binding and release of the substrate occurs on the cytoplasmic side. The structural pseudo-symmetry found within each protomer[20], led initially to the hypothesis that hairpin 1 (HP1), which is pseudo-symmetry-related to HP2 would hinge and act as a gate at the cytoplasmic side. As in such a transport mechanism the substrate would pass through two different structural elements (HP1 and HP2), it is referred to as the two-gate elevator mechanism. Alternatively, the possibility that HP2 might also open from the intracellular side (one-gate elevator mechanism) has been discussed[2,21]. A definite answer to this key mechanistic question can only be resolved by the structure of an inward-open state of an SLC1A transporter. Here, we present cryo-EM structures of ASCT2 with a single mutation in the binding site (ASCT2_C467R) in the presence and absence of its competitive inhibitor TBOA, and of the substrate free wild-type ASCT2_wt in the presence of Na+. All structures capture an inward-open state, where HP2 has moved indicating a one-gate elevator mechanism.

## Results

**Altering the substrate specificity of ASCT2.** We recently reported a cryo-EM structure of the human ASCT2 in a glutamine-bound inward-occluded state[2]. We reckoned that binding of bulkier substrates might prevent full occlusion and capture the protein in an inward-open state—a strategy previously used to obtain the outward-open structures of Glt_Ph and EAAT1, where binding of DL-TBOA and TFB-TBOA prevented closure of HP2[15,19]. However, as ASCT2 does not transport aspartate, these bulky aspartate analogues do not inhibit its transport activity (Fig. 1a). A comparison of the substrate-binding sites in the ASCT2[2] and the EAAT1[19] structures indicates that the difference in substrate and inhibitor specificity might be largely due to a single amino acid substitution, namely Cys467 in ASCT2 replacing Arg459 in EAAT1. It appears that the arginine residue is the main determinant for acidic amino acid specificity, as the guanidium group interacts with the side chain carboxylate of glutamate and aspartate in transporters as Glt_Ph, Glt_Tk and EAAT1. A substitution to a cysteine in EAAT3 has been shown to abolish acidic amino-acid transport and instead introduced neutral amino-acid specificity[22]. Based on this data, we expected that the substitution of Cys467 in ASCT2 to arginine might change its substrate specificity to acidic amino acids and would allow inhibition by TBOA. We engineered this mutation in human ASCT2 (ASCT2_C467R), purified the protein

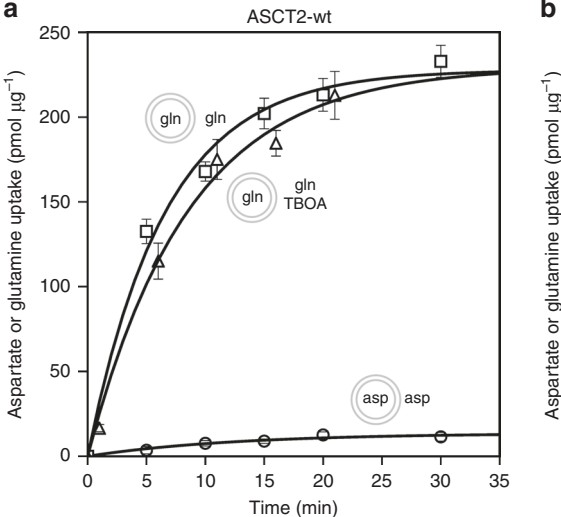

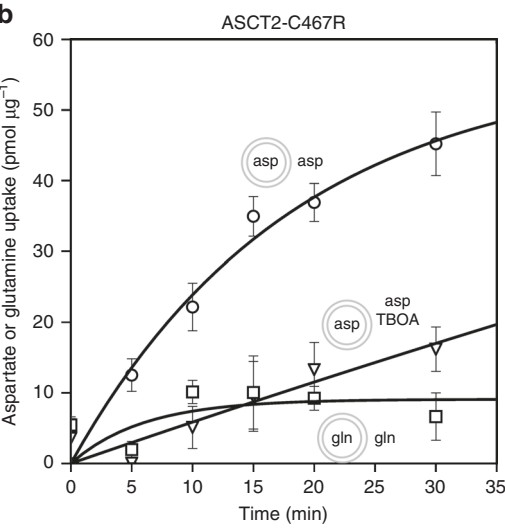

**Fig. 1** Transport activity of reconstituted purified ASCT2_wt (**a**) and ASCT2_C467R (**b**). Curves show the counterflow of external radioactively-labelled and internal unlabelled ʟ-glutamine (squares, $n = 6$ for ASCT2_wt, $n = 3$ for ASCT2_C467R); counterflow of ʟ-glutamine in presence of 0.6 mM DL-TBOA (upward triangle, $n = 4$); counterflow of ʟ-aspartate (circles, $n = 3$ for ASCT2_wt, $n = 5$ for ASCT2_C467R); and counterflow of ʟ-aspartate in presence of 0.5 mM DL-TBOA (downward triangles, $n = 3$). Data points and error bars represent means and s.e.m. from $n$ biologically independent experiments. Small schemes depict proteoliposomes with internal and external compound compositions used in respective experiments. Source data are provided as a Source Data file

from *Pichia pastoris* and monitored its substrate uptake in proteoliposomes. Indeed, the substrate specificity of ASCT2_{C467R} differed from that of the wild-type. ASCT2_{C467R} did no longer support glutamine uptake, but it transported aspartate instead (Fig. 1b). Although the uptake rate of aspartate by ASCT2_{C467R} was slower than that of glutamine by ASCT2_{wt}, the results confirm that a single point mutation in the substrate binding site was sufficient to switch the substrate specificity of ASCT2 from neutral to acidic amino acids. In addition, aspartate transport could be suppressed by TBOA (Fig. 1b), which opened the possibility to use the compound to trap the transporter in an open state.

**Cryo-EM structures of inward-open ASCT2**. We determined the structure of the 172-kDa trimeric ASCT2_{C467R} in presence of TBOA at 3.6 Å resolution by single particle cryo-EM (Fig. 2, Supplementary Figs. 1 and 4, Table 1). The protein was embedded in *n*-dodecyl-β-d-maltopyranoside (DDM) micelles supplemented with cholesterol hemisuccinate (CHS), which has been shown to increase stability[2]. The quality of the cryo-EM map was sufficient to unambiguously model residues 43–159 and 177–489. Notably, despite different strategies employed during image processing, including 3D classifications with and without symmetry imposed on the trimer or single protomers, only a single conformational state was captured for all protomers, indicating a preferred state of the transporter in the given conditions (Supplementary Fig. 1d). Strikingly, the ASCT2_{C467R} structure reveals an inward-open state of a SLC1A member (Figs. 3 and 4).

Compared to the previously determined substrate-bound inward-occluded state of ASCT2[2], the most prominent difference is the conformation of the loop connecting both helices of hairpin 2 (HP2 loop). It has hinged away from the glutamine-binding site towards the scaffold domain by ~8 Å, opening a pathway to the cytoplasm (Figs. 3a and Fig. 4a, b, Supplementary Movie 1). Similar movements of the HP2 loop have been observed on the extracellular side between the outward-occluded and outward-open states of EAAT1 and Glt_Ph structures[15,19]. Although the remainder of the transport domain has virtually the same conformation as in the inward-occluded state (Fig. 3a), it is slightly further tilted towards the cytoplasm, away from the scaffold domain providing additional space for HP2 to open (Supplementary Fig. 5a, Supplementary Movie 1).

In the substrate-binding site we observe a patch of non-protein density close to R467. Although the density is too weak to allow for an unambiguous assignment, it most likely represents a TBOA molecule, although the occupancy of the site appears low (Supplementary Fig. 5b). To test whether TBOA binding was required to arrest the protein in the inward-open state, we further determined a cryo-EM structure of ASCT2_{C467R} in the absence of its inhibitor. A 4.1 Å resolution cryo-EM map of substrate-free ASCT2_{C467R} shows a virtually identical inward-open state with a displaced HP2 loop (Supplementary Figs. 2, 4 and 6a–c, Table 1). This is surprising, as the binding of bulky inhibitors was required to obtain all of the reported outward-open states of EAAT1 and Glt_Ph[15,19,23]. To exclude that the C467R mutation itself induced the observed opening of HP2, we also investigated the conformational state of ASCT2_{wt} in absence of its substrate

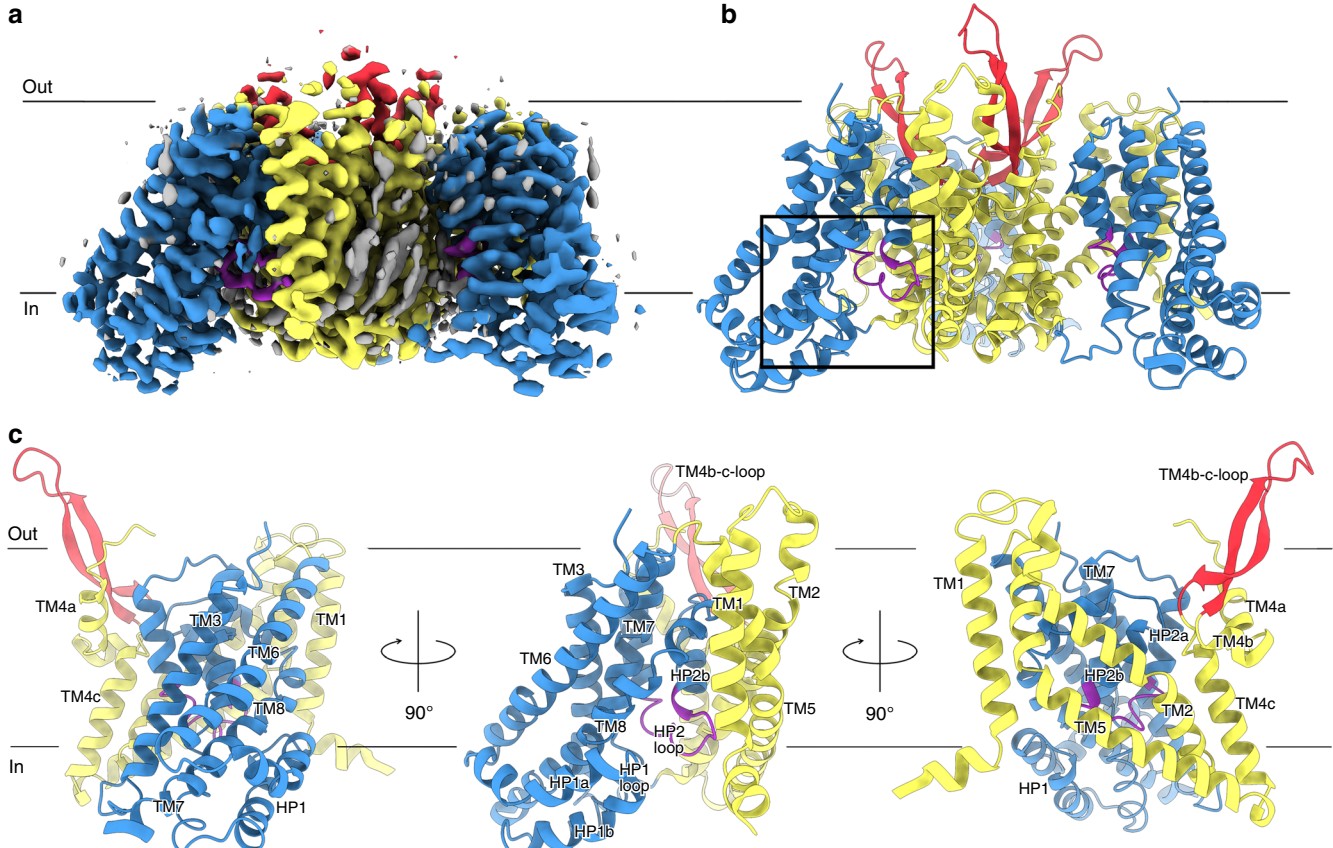

**Fig. 2** Inward-open structure of ASCT2_{C467R}. Cryo-EM map of ASCT2_{C467R} obtained in presence of TBOA at 3.6 Å resolution (**a**) and respective model shown as ribbon (**b**). **c** Protomer of ASCT2 with labelled structural elements is shown as ribbon, rotations are indicated. The scaffold domains are coloured in yellow, the extracellular antenna in red, the transport domains in blue, the HP2 loop in purple, and non-protein densities in grey (see also Supplementary Fig. 7). The membrane boundary is indicated and the location of the substrate binding site as shown in Fig. 3 is highlighted by a black rectangle in panel b

**Table 1 Cryo-EM data collection, refinement and validation statistics**

| | ASCT2$_{C467R}$—TBOA (EMD-10016, PDB 6RVX) | substrate-free ASCT2$_{C467R}$ (EMD-10018, PDB 6RVY) | substrate-free ASCT2$_{wt}$ (EMD-10017) |
|---|---|---|---|
| *Data collection and processing* | | | |
| Magnification | 49,407 | 49,407 | 49,407 |
| Voltage (kV) | 200 | 200 | 200 |
| Electron exposure (e−/Å$^2$) | 52 | 52 | 52 |
| Defocus range (μm) | −0.8 to −1.8 | −0.8 to −1.8 | −0.8 to −1.8 |
| Pixel size (Å) | 1.012 | 1.012 | 1.012 |
| Symmetry imposed | C3 | C3 | C3 |
| Initial particle images (no.) | 987,852 | 245,367 | 317,611 |
| Final particle images (no.) | 223,354 | 23,664 | 19,651 |
| Map resolution (Å) | 3.61 | 4.13 | 6.98 |
| FSC threshold | 0.143 | 0.143 | 0.143 |
| Map local resolution range (Å) | 4.2–3.4 | 6.4–4.1 | 8.0–6.3 |
| *Refinement* | | | |
| Initial model used | PDB 6GCT | PDB 6GCT | – |
| Model resolution (Å) | 3.6 | 4.1 | – |
| (0.5 FSC threshold) | | | |
| Model resolution range (Å) | 15–3.6 | 15–4.1 | – |
| Map sharpening *B* factor (Å$^2$) | −205 | −211 | −764 |
| Model composition | | | |
| Nonhydrogen atoms | 9627 | 9627 | – |
| Protein residues | 1290 | 1290 | – |
| Ligands | 0 | 0 | – |
| *B* factors (Å$^2$) | | | |
| Protein | 19.03 | 83.05 | – |
| Ligand | – | – | – |
| R.m.s. deviations | | | |
| Bond lengths (Å) | 0.006 | 0.006 | – |
| Bond angles (°) | 0.979 | 1.023 | – |
| Validation | | | |
| MolProbity score | 1.75 | 1.84 | – |
| Clashscore | 5.00 | 6.72 | – |
| Poor rotamers (%) | 0.29 | 0.49 | – |
| Ramachandran plot | | | |
| Favoured (%) | 92.02 | 92.49 | – |
| Allowed (%) | 7.98 | 7.51 | – |
| Disallowed (%) | 0.00 | 0.00 | – |

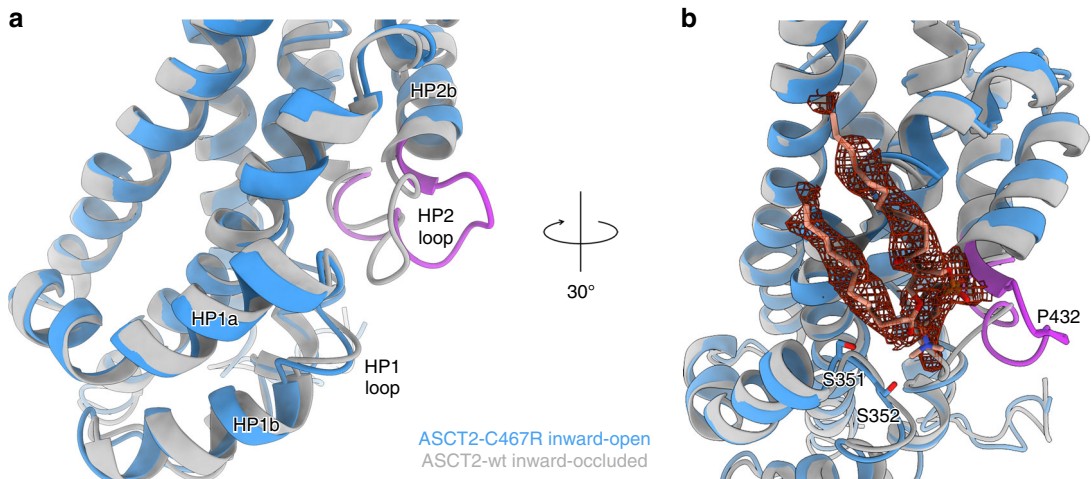

**Fig. 3** Substrate-binding site and movement of the HP2 loop. **a**, **b** Superposition of the inward-open ASCT2$_{C467R}$ in presence of TBOA (shown in blue with the HP2 loop in purple) and the substrate-bound inward-occluded ASCT2$_{wt}$ (shown in grey, PDB-ID: 6GCT), demonstrating the movement of HP2. **b** Non-protein density (shown as red mesh at 3σ) located near HP1 and HP2 with a modelled diundecylphosphidylcholine and putatively coordinating residues shown as sticks. The lipid head group is positioned between the HP1 and HP2 loops and its negative charge appears to be stabilised by the helix dipole of hairpin helix HP2b. **a**, **b** Structures were superimposed on the transport domain

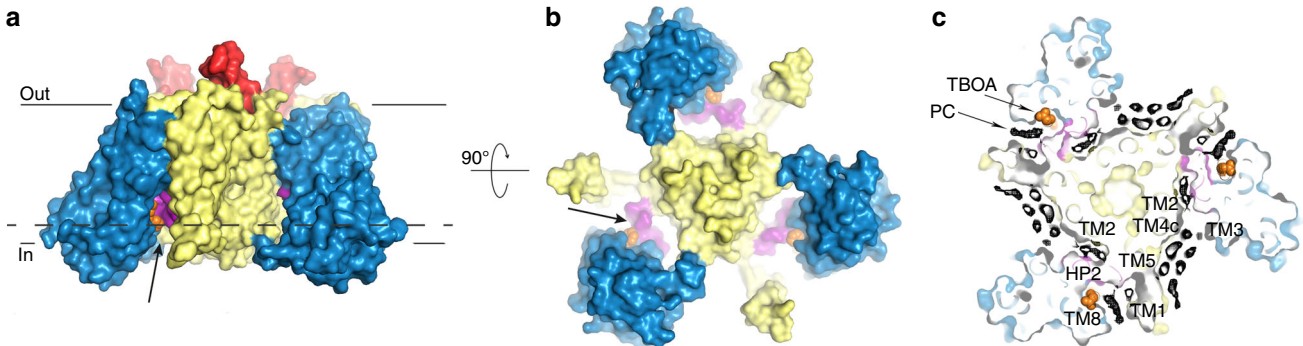

**Fig. 4** Accessibility of the substrate-binding site and lipid densities. **a**, **b** Surface representation of the ASCT2$_{C467R}$ structure in presence of TBOA, with the transport domain in blue, the scaffold in yellow, the antennae in red, the HP2 loop in purple and TBOA in orange. The open HP2 gate provides access to the binding site from the cytoplasm (indicated by arrows). **c** Slice through the surface representation of ASCT2$_{C467R}$ (level is indicated by dashed line in panel **a**) with putative lipid densities surrounding the scaffold domain shown in black and TBOA as orange balls (see also Supplementary Fig. 7)

glutamine. In the absence of substrate ASCT2$_{wt}$ appears to be less stable and the obtained data suffered from lower particle quality compared to ASCT2$_{C467R}$. Albeit of lower resolution, the cryo-EM map of the substrate-free ASCT2$_{wt}$ shows that the protein accommodates the same inward-open state, where the HP2 loop has moved (Supplementary Figs. 3 and 6d–f, Table 1). It appears that an inward-open state of ASCT2 is favoured in the given conditions without its respective substrate (L-glutamine in ASCT2$_{wt}$) added. Thus, neither the presence of the inhibitor nor the point mutation itself is responsible for gate opening on the intracellular side, allowing the use of the higher resolved ASCT2$_{C467R}$ cryo-EM map for a detailed description of the inward-open state.

**Potential allosteric binding sites**. In the cryo-EM maps of the inward-open states of ASCT2$_{C467R}$, an additional density is present between the HP1 and HP2 tips (Figs. 3b and 4c). The characteristic shape of this density indicates that it most likely corresponds to a lipid molecule. The exact lipid identity is ambiguous at the given resolution, but the tentatively modelled diundecyl phosphatidylcholine fits well in the density (Fig. 3b). Notably, the head group occupies the position where the HP2 loop is located in the inward-occluded state, indicating that the lipid would need to relocate for gate closure (Fig. 3). Consistently, no lipid density was observed in the substrate-bound inward-occluded ASCT2$_{wt}$ structure[2].

Further unassigned densities were found between the transport and scaffold domains (Figs. 2a and 4c and Supplementary Fig. 7). The density at the interface between HP2 and TM2 of the scaffold domain is located near the putatively bound cholesterols found in the inward-occluded ASCT2 structure[2] (Fig. 4c, Supplementary Fig. 7a–c). However, as the position of HP2 in the inward-open structure partly overlaps with the potential lipid densities found in the inward-occluded structure, the previously assigned lipids must relocate upon HP2 opening. Another patch of unassigned density is located between the HP2 loop and TM1 and TM5 (Fig. 4c, Supplementary Fig. 7a, b, d, e). This putative lipid is adjacent to the phosphatidylcholine found at the tip of both hairpins described above (Figs. 3b and 4c, Supplementary Fig. 7a, b, d, e). It might be responsible for the observed helix-rearrangement of TM1 of the scaffold domain, which is not directly interacting with HP2, when compared to the inward-occluded ASCT2 structure[2] (Supplementary Fig. 7e, Supplementary Movie 1). A large group of well defined, unassigned densities, which likely represent lipids or cholesterol molecules, is found in the cavity formed by TM5 of one protomer, and TM2-4 and the back of HP2 of the adjacent protomer (Fig. 4c, Supplementary

Fig. 7a, b, f). Interestingly, one of these densities is located at a similar region as where the allosteric inhibitor UCPH$_{101}$ binds EAAT1[19], hinting at a potential regulatory role. Finally, a density was found on the extracellular side of the scaffold domain. It is located between the N- and C-termini of the ASCT2 antennae (TM4b,c-loop) of two adjacent protomers (Supplementary Fig. 7a, g), with distances to neighbouring side chains of ~4 Å. The shape of the density suggests that it might represent a hydrated bound ion, but it is not possible to unambiguously determine the identity from the density map.

## Discussion
The cryo-EM structures of the inhibitor-bound ASCT2$_{C467R}$, the substrate-free ASCT2$_{C467R}$ and the substrate-free ASCT2$_{wt}$ presented here, reveal the hitherto elusive inward-open conformation of an SLC1A transporter. They show how HP2, and not HP1, opens to provide access for binding and release of the substrate on the cytoplasmic side of the membrane. Thus, ASCT2 functions as a one-gate elevator, where the HP2 loop serves as a gate on both the extracellular and intracellular sides of the membrane (Fig. 5, Supplementary Movie 1). Given the structural similarity between ASCT2 and other SCL1A members, it is likely that the same structural element constitutes the gate throughout the family. These findings broaden our general understanding of the transport mechanisms of secondary-active transporters. Unique for SLC1A transporters, is that substrate binding and release is gated via the same structural element, the substrate is fully occluded inside the transport domain, which then moves in an elevator-like mechanism from one side of the membrane to the other (one-gate elevator mechanism). This contrasts, e.g., with the well-characterised transport mechanisms of transporters belonging to the major facilitator family (MFS) or the LeuT-fold family[24,25]. In these families, the substrate becomes sandwiched between two lobes or domains during the translocation, whereby a rocking movement exposes different structural elements to the extra- and intracellular sides of the membrane to gate the access to the binding site (two-gate rocking mechanism).

Further, we were able to identify several non-protein densities that likely correspond to lipid or cholesterol molecules. Putative lipid molecules that directly and indirectly interact with the HP2 gate in the inward-open state (Figs. 2a, 3b and 4c, Supplementary Fig. 7) will have to relocate to allow the transition to a substrate-bound inward-occluded state. We thus predict that these sites are potential allosteric binding sites that might affect the dynamics of the transport cycle. The identification of these sites could guide the development of novel lipid-based drugs for cancer treatment. In particular the well-defined site for a co-purified lipid from

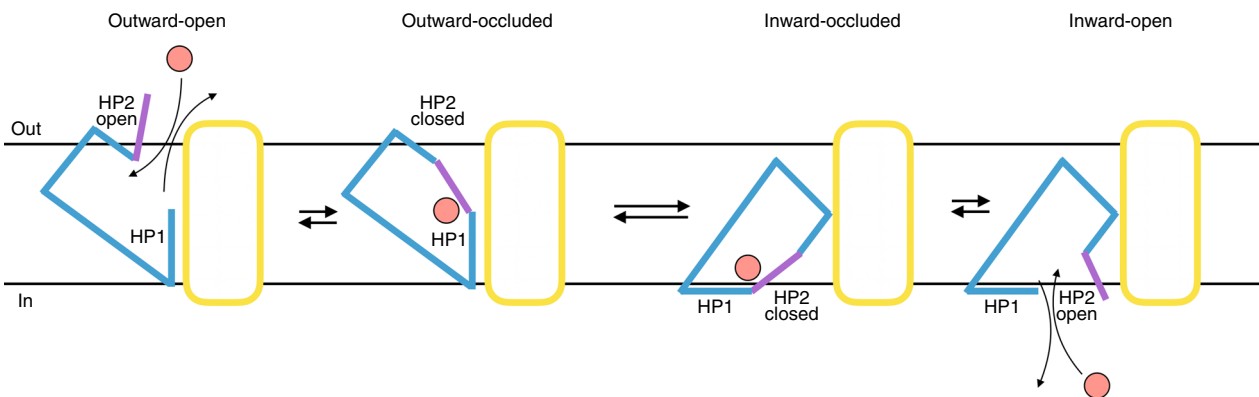

**Fig. 5** One-gate elevator transport mechanism. Schematic representation of the transport cycle of SLC1A transporters, with the scaffold domain in yellow, the transport domain in blue, the extra- and intracellular gate (HP2 loop) in purple and the transported amino acid in pink

*Pichia pastoris* found between the hairpins HP1 and HP2 (Fig. 3b) could be a potential target location for the development of binders that could block ASCT2 activity and prevent glutamine supply in cancer cells. Such lipid-like binders could act as inhibitor, arresting the protein in the observed inward-open state of ASCT2.

In conclusion, in light of the pharmacological relevance of ASCT2 as a target in anticancer therapy, we anticipate that the structure and mechanistic insights provided in this study may offer new entry points for rational drug design.

## Methods

**ASCT2$_{C467R}$ engineering**. The C467R single-point mutation was introduced using QuickChange site-directed mutagenesis (forward primer 5′ GGTTGACAGATCTA GAACCGTCTTGAACGTTG 3′; reverse primer 5′ CAACGTTCAAGACGGTTC TAGATCTGTCAACC 3′) on the pPICZ-A plasmid, containing the codon optimised human ASCT2 gene[26]. The obtained construct was sequenced (forward primer 5′ GCGACTGGTTCCAATTGACAAGC 3′; reverse primer 5′ CAAATGG CATTCTGACATCCTCTTG 3′), linearised and transformed into the *Pichia pastoris* X-33 strain by electroporation (Invitrogen). Colony selection was performed on YPDS plates with increasing concentration of Zeocin (ThermoFisher Scientific), namely 100, 1000 and 2000 μg/ml. Colonies from plates with the highest Zeocin concentrations were further used for small-scale expression tests and the amount of expressed ASCT2 was estimated by Western blot using antibodies against the His$_4$-tag (QIAGEN). ASCT2$_{wt}$ colonies grown at equivalent conditions were used as positive control to compare protein expression levels. For screening, 10 ml of BMGY medium were inoculated with a colony and incubated overnight at 30 °C in a shaker at 200 rpm. Next morning cells were centrifuged at room temperature (21 °C) for 10 min at 3000×g, resuspended in 10 ml of BMMY (1% methanol) medium and grown for 24 h at 30 °C in a shaker at 200 rpm. Cells were subsequently centrifuged at 4 °C for 10 min at 3000×g, washed with 10 ml of buffer A (20 mM Tris-HCl pH 7.5, 100 mM NaCl, 0.5 mM EDTA, 5% w/v glycerol) and resuspended in 500 μl lysis buffer (buffer A + 1 mM PMSF and 0.1 mg ml$^{-1}$ DNase A). Approximately 500 μl glass beads were added and cells were lysed using the TissueLyser II (QIAGEN) for 5 min at 50 Hz. The lysed cells were centrifuged at 4 °C for 5 min at 11,700×g, an additional 200 μl of lysis buffer was added to the cells and the sample was centrifuged. The membrane fraction was obtained by centrifugation of the supernatant for 1 h at 4 °C, 20,817×g, resuspended in 40 μl buffer A, flash-frozen in liquid nitrogen and stored at −80 °C.

**Protein expression and purification**. Large-scale ASCT2$_{C467R}$ protein overexpression was carried out using baffled flasks[27], whereas for ASCT2$_{wt}$ overproduction in a fermentor was used for cell cultivation[26]. Membrane fraction was collected by ultracentrifugation of the supernatant (120 min, 193,727 g, 4 °C) after breaking cells with a Constant Cell Disruption System (three passages at 39 kPsi, 5 °C)[2]. To purify ASCT2, an aliquot of membrane vesicles representing ~1.5 g cells was solubilized in buffer B (25 mM Tris-HCl, pH 7.4, 300 mM NaCl, 10% (vol/vol) glycerol, 1% DDM and 0.1% CHS (Anatrace)) for 1 h at 4 °C. After ultracentrifugation (30 min, 442,907×g, 4 °C) the supernatant was incubated with Ni$^{2+}$-Sepharose resin for 1 h at 4 °C. The column was washed with buffer C (20 mM Tris-HCl, pH 7.4, 300 mM NaCl, 50 mM imidazole, pH 7.4, 10% glycerol, 0.05% DDM and 0.005% CHS), and protein was eluted with buffer C containing 500 mM imidazole. Peak elution fraction was applied to size-exclusion chromatography with a Superdex 200 10/300 gel-filtration column (GE Healthcare)

preequilibrated with buffer D (20 mM Tris-HCl, pH 7.4, 300 mM NaCl, 0.05% DDM and 0.005% CHS).

**Reconstitution into proteoliposomes and transport assays**. Freshly purified ASCT2 was used for reconstitution into proteoliposomes as previously described in presence of the indicated amino acid substrate[2]. The liposomes were composed of *Escherichia coli* polar lipids and egg phosphatidylcholine at a 3:1 ratio (w/w) and supplemented with 10% (w/w) cholesterol (Avanti Polar Lipids). For transport assays with [$^3$H] glutamine and [$^3$H]aspartate (PerkinElmer), proteoliposomes were loaded with 50 mM NaCl and 10 mM glutamine/aspartate using three freeze-thawing cycles, then extruded 11 times through a 400-nm-diameter polycarbonate filter (Avestin), diluted in buffer E (20 mM Tris pH 7.0) and ultracentrifuged for 45 min at 4 °C with 442,907×g. Proteoliposomes were resuspended in minimal volume of buffer E (~1 μg protein per 1.5 μl). The transport assay was carried out in a water bath at 25 °C with constant stirring. The external substrate mixture (50 mM NaCl and 50 μM [$^3$H]glutamine/[$^3$H]aspartate) was pre-warmed to the temperature of the assay. Transport was initiated by dilution of 1.5 μl proteoliposomes in 80 μl external buffer. At indicated time points the reaction was stopped by diluting the mixture in 2 ml of cold buffer E, filtered over a 0.45-μm pore-size filter (Portran BA-85, Whatman), washed with 2 ml of cold buffer E and filtered again. The level of radioactivity accumulated inside the proteoliposomes, as a consequence of amino-acid exchange, was counted using a PerkinElmer Tri-Carb 2800RT liquid scintillation counter after dissolving the filter in 2 ml of scintillation liquid (Emulsifier Scintillator Plus, PerkinElmer).

**Cryo-EM sample preparation and data collection**. Freshly purified ASCT2$_{C467R}$ (in mixed DDM + CHS micelles) was concentrated to ~5 mg ml$^{-1}$ using an Amicon Ultra-0.5 mL concentrating device (Merck) with a 100 kDa filter cut-off. For the cryo-EM data collection in presence of inhibitor, DL-threo-β-benzyloxyaspartic acid (DL-TBOA) (Hello Bio) was added to the protein sample to a final concentration of 1 mM prior to sample freezing. The ASCT2$_{wt}$ sample was freshly purified in absence of amino acid substrate and concentrated to ~2 and ~8 mg ml$^{-1}$. Holey-carbon cryo-EM grids (Au R1.2/1.3, 300 mesh, Quantifoil) were glow discharged at 5 mA for 20 s. 2.8 μl protein sample were applied onto the grids, blotted for 3–5 s in a Vitrobot Mark IV (Thermo Fisher) at 15 °C and 100% humidity, plunge frozen into a liquid ethane/propane mixture and stored in liquid nitrogen until further use. Cryo-EM data were collected in-house on a 200-keV Talos Arctica microscope (Thermo Fisher) with a post-column energy filter (Gatan) in zero-loss mode, with a 20-eV slit and a 100-μm objective aperture. Prior to data collection, the best regions on the grid were identified with the help of a self-written script to calculate the ice thickness (manuscript in preparation). Images were acquired in an automatic manner with EPU (Thermo Fisher) on a K2 summit detector (Gatan) in counting mode at ×49,407 magnification (1.012 Å pixel size) and a defocus range from −0.8 to −1.8 μm. During an exposure time of 9 s, 60 frames were recorded with a total dose of about 52 electrons/Å$^2$. During data collection the FOCUS software[28] was used to monitor data quality on-the-fly and to adjust settings if necessary.

**Image processing**. For the ASCT2$_{C467R}$ dataset in presence of TBOA, 7327 micrographs were recorded. Beam-induced motion was corrected with Motion-Cor2_1.1.0[29] and the CTF parameters estimated with ctffind4.1.8[30]. Manual selection of micrographs was carried out in FOCUS, whereby micrographs out of defocus range (<0.4 and >2 μm), contaminated with ice or aggregates, and with a low-resolution estimation of the CTF fit (>5 Å), were discarded. The remaining 6698 micrographs were imported in RELION-3.0-beta-2[31] for further image processing. Around 1200 particles were manually picked to calculate an initial template for autopicking. After parameter optimisation 1,109,646 particles were autopicked and extracted with a box size of 240 pixels. Several rounds of 2D and

3D classifications allowed to select 375,596 high quality particles. The final set of particles was determined in a last round of 3D classification with 6 classes, of which the best particles clustered in one class with 223,354 particles (59.5%). These particles were subjected to auto-refinement, yielding a map of 4.12 Å. In the last refinement iteration, a mask excluding the micelle was used and the refinement continued until convergence (focus refinement), which improved the resolution to 3.92 Å. During classification and refinement, a C3 symmetry was imposed. Initially, the structure of ASCT2$_{wt}$[2] was used as a reference and in an iterative way, the new obtained best map was used in subsequent rounds. References were low-pass filtered to 40 or 50 Å. The final map was masked and sharpened during post-processing resulting in a resolution of 3.68 Å. Finally, the newly available algorithm of per-particle CTF refinement and beam tilt refinement in Relion3[31] was used on re-extracted particles with a box size of 200 pixels to further improve the resolution to 3.61 Å. Subtraction of the detergent micelle did not improve resolution. 3D classifications without imposed symmetry as well as on individual protomers with symmetry expansion and signal subtraction were performed at different levels of image processing to check for conformational heterogeneity between and within ASCT2 trimers. Local resolution estimations were determined in Relion. All resolutions were estimated using the 0.143 cut-off criterion[32] with gold-standard Fourier shell correlation (FSC) between two independently refined half-maps[33]. During post-processing, the approach of high-resolution noise substitution was used to correct for convolution effects of real-space masking on the FSC curve[34]. The directional resolution anisotropy of density maps was quantitatively evaluated using the 3DFSC web interface (https://3dfsc.salk.edu)[35].

A similar approach was performed for the image processing of the ASCT2$_{C467R}$ dataset in absence of TBOA. In short, 1605 micrographs were recorded, and 1479 used for image processing after selection. Over 323,403 particles were autopicked and subjected to several rounds of 2D and 3D classification. The final data comprising 23,367 particles yielded after refinement and masking a map of 4.13 Å resolution.

For the dataset of ASCT2$_{wt}$ in absence of substrate, in total 4305 micrographs were recorded and 3272 micrographs were used for further image processing. After manual particle picking, over 450,000 particles were autopicked and subjected to several rounds of 2D and 3D classification. The final data comprising 19,651 particles yielded after refinement and masking a map of 6.98 Å resolution. 3D classifications without imposed symmetry performed at different stages of image processing, as well as a 3D classification of individual protomers after symmetry expansion and signal subtraction, yielded the same inward-open conformation of the protein.

**Model building**. Model building was carried out in COOT[36] using the previously determined ASCT2$_{wt}$ structure[2] as reference. The resolution of the map was of sufficient quality to unambiguously assign the protein sequence and model most of the residues (43–159, 177–489). After each round of real-space refinement performed in Phenix[37] with NCS restrains, coordinates were manually inspected and edited in COOT and submitted to another refinement round in an iterative way. The quality of the fit was validated by a Fourier shell cross correlation (FSC$_{sum}$) between the refined model and the final map. To monitor the effects of potential overfitting, random shifts (up to 0.5 Å) were introduced into the coordinates of the final model, followed by refinement against the first unfiltered half map. The FSC between this shaken-refined model and the first half map used during validation refinement is termed FSC$_{work}$, and the FSC against the second half map, which was not used at any point during refinement, is termed FSC$_{free}$. A marginal gap between the curves describing FSC$_{work}$ and FSC$_{free}$ indicates no overfitting of the model. The chemo-physical properties of the final model were validated with MolProbity[38].

The SBGrid software package tool was used[39]. Images were prepared with PyMOL (The PyMOL Molecular Graphics System, Version 2.0 Schrödinger, LLC), ChimeraX[40], and a movie was made with Chimera[41].

**Reporting summary**. Further information on research design is available in the Nature Research Reporting Summary linked to this article.

## Data availability

Data supporting the findings of this manuscript are available from the corresponding authors upon reasonable request. A reporting summary for this Article is available as a Supplementary Information file.

The three-dimensional cryo-EM density maps of the inward-open ASCT2$_{C467R}$ in presence of TBOA, the substrate-free ASCT2$_{C467R}$ and the substrate-free ASCT2$_{wt}$ have been deposited in the Electron Microscopy Data Bank under accession numbers EMD-10016, EMD-10018 and EMD-10017, respectively. The deposition includes the cryo-EM maps, both half-maps, the unmasked and unsharpened refined maps and the mask used for final FSC calculation. Coordinates of two models have been deposited in the Protein Data Bank. The accession numbers for ASCT2$_{C467R}$ in presence of TBOA and substrate-free ASCT2$_{C467R}$ are 6RVX and 6RVY, respectively. The source data underlying Fig. 1 are provided as a Source Data file.

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

## Acknowledgements

We thank G.T. Oostergetel for help with cryo-EM data acquisition; M. Punter for maintenance of the image-processing cluster; R.C. Prins for help with uptake measurements. This research was supported by NWO Vidi grant 723.014.002 to A.G.; NWO Vici grant 865.11.001 and European Research Council Starting Grant 282083 to D.J.S; and the NWO Veni grant 722.017.001 and the NWO Start-Up grant 740.018.016 to C.P.

## Author contributions

All authors designed experiments, A.A.G performed all experiments. All authors analysed and interpreted the data and contributed to writing the manuscript.

## Additional information

**Competing interests:** The authors declare no competing interests.

