## [Peer Review File · Nature Communications]

Reviewers' Comments:

Reviewer #1:

Remarks to the Author:

Garaeva, Guskov, Slotboom and Paulino present excellent results on the inward-open conformation of the neutral amino acid transporter ASCT2, which is of the same protein family as the glutamate transporters of the EAAT family. A good 3.6 Å resolution cryo-EM structure is obtained for a Cys367Arg mutation (mimicking the substrate site of EAAT's and now gaining Asp transport while losing Gln transport) with negatively charged TBOA inhibitor and a lower resolution structure is obtained for the wt enzyme, showing the same inward-open state. The authors find the HP1 loop to gate also the inward oriented state, as it does for the outward-oriented states. They conclude that these transporters use the combination of an elevator mechanism and a single gate. This is very interesting from a mechanistic point of view for EAAT's specifically and transporters in general.

The cryo-EM and functional studies in liposomes is beautifully done and provides an excellent basis for the report, and I have only manageable comments on the report:

Overall, the report is very short and could gain from being expanded a bit. For example to include a (much) closer comparison to the inward-open structure obtained by Hg crosslinking (Reyes et al., ref. 16).

Also the final remarks on drug discovery using this mechanistic insight is very vague - please be more specific or end on a different note.

small comment:

Line 27: please explain "40% similarity" - this is an unclear term

Please

Reviewer #2:

Remarks to the Author:

Review:

Summary:

Garaeva et al. present three structures of the Alanine Serine Cysteine transporter 2 (ASCT2) in the inward open state – two of a point mutant, C467R, in the presence and absence of ligand, and one of the wild-type protein in the apo state. The best of these, the C467-TBOA structure, is resolved to 3.6Å. All three structures are very similar in conformation, and differ from the inward occluded structure of the same protein solved by Dr. Paulino's lab primarily in the movement of a loop, HP2, which occludes the ligand binding site in the previous structure, while exposing it in all three structures presented in this manuscript. This is significant, because this is the same loop that also acts as a gate to entry of the ligand in the outward-open structures of other members of the SLC1A family, demonstrating that SLC1A transporters operate via a one-gate elevator mechanism, rather than a two-gate mechanism as previously theorized. In addition to the structural data, the authors also present functional assays demonstrating that a single point mutation, C467R, is sufficient to switch specificity of the transporter from neutral amino acids to acidic substrates.

Main Impressions:

This is an excellent and thorough study, with high quality structural data addressing a clearly posed question – how do SLC1A transporters release substrates into the cell after transport across the membrane? Having previously solved the structure of an inward occluded form of the same transporter (wt-ASCT2 bound to glutamine), the authors were in an excellent position to resolve this question. The authors first attempted to use a bulky substrate (TBOA) to preclude the inner

gate from closing, trapping the transporter in the inward open form. This approach had been previously used to capture the outward-open state of another SLC1A transporter, EAAT1. However, this substrate did not appreciably inhibit transport of glutamine by ASCT2, which the authors attributed to the specificity of ASCT2 for neutral rather than acidic substrates. To address this, Garaeva et. al. introduced a point mutation in the ligand binding pocket (C467R) intended to convert ASCT2 to a transporter for acidic rather than neutral amino acids, and also make the transporter amenable to inhibition by TBOA. This approach was successful (the mutant receptor now transports aspartate preferentially to glutamine and is inhibited by TBOA), although it appears to be much less active overall than the wild type protein. The authors go on to solve the structure of ASCT2-C467R in the presence of TBOA at 3.6Å, and the density map and model are overall of excellent quality (see minor comments below). The density for the ligand however is very poor, and it is not clear to me based on the density that it is bound at the assigned location with any appreciable occupancy. However, ASCT2 is clearly in the inward-open state, based on comparison of the map and model with the inward occluded state of the protein – the movement of the HP2 loop to uncover the ligand binding cavity is very clear. In order to test whether the added ligand is indeed having the expected effect, and whether the introduced mutation is affecting the conformation of HP2, the authors solved two additional, lower resolution structures, of C467R in the absence of ligand, and of the wild-type protein in the apo state. Both of these structures look identical to the notionally ligand-bound mutant structure. Given the absence of ordered density for the TBOA molecule, this suggests that the ligand was perhaps not bound in the initial dataset, or at least that there is little evidence for TBOA binding in the current study. This is not of critical importance however, as the protein is clearly in the inward-open configuration in all three states, and in all three states opening is mediated by movement of HP2, which answers the question that the authors set out to address. I do feel the apparently unexpected finding that the inner gate is open even in the absence of added inhibitor could do with a little further discussion. Overall, while I have a few minor technical comments below that I would like to see addressed, this is an excellent study that I am happy to recommend for publication.

Signed: Oliver B Clarke, Columbia University.

Specific comments:

Image processing

Overall, the image processing workflow is very clear and easy to follow – Supp Figs1-4 could serve as a guide for others in the field in how to present this information in an easily digestible fashion! I particularly appreciate that the mask used in FSC calculation is depicted on the FSC curve, and that the per-TM density figures have been calculated in a manner that does not recontour the density to close over holes in the mesh (which is quite common when using Pymol). I have a couple of minor queries:

- It is stated that the map was obtained using C3 symmetry, but the supplied map seems to have some differences between the three protomers – mostly in the distribution of noise peaks. This should not be the case if symmetry was enforced. Perhaps some procedure after refinement (e.g. local resolution filtering) introduced these features?
- In the text, the global resolution is listed as 3.61 Å, but in Table 1 the resolution range for the map is listed as 4.2-3.4. To be clear, the map has been filtered to 3.6Å, correct? If this is meant to be the local resolution range, I would suggest specifying it as such (as opposed to being the resolution range of spatial frequencies included in the final map).
- CTF refinement in Relion is mentioned, but further details should be supplied for completeness – was this just per particle defocus refinement? Beam tilt refinement?

Model building and ligand assignment

- There is very little support in the density map for the assignment of the TBOA ligand – there is a small peak where the ligand has been assigned, but this peak is smaller than many of the noise peaks in the micellar region, and much weaker than any of the unassigned lipid/cholesterol/detergent densities. I think it is worth mentioning the presence of this peak, but

without stronger evidence (e.g. a difference density with an apo map of comparable resolution), I don't think it is justifiable to include the ligand in the deposited model. In addition, the ligand appears to be assigned as covalently bound to R467, with a distance of 1.3Å from O4 of TBOA to NE of R467. Is this intended/expected?

- There are several cis-peptide bonds of non-Pro residues which should be fixed – check residues 122, 124, 428 & 367. 428 in particular needs attention, as it is located in HP2, near the ligand binding site.
- The conformation of HP2 differs between the three protomers – it is more or less concordant with the density in chain A (except for the cis peptide at A428 mentioned above), but in chains B and C P432 and I431 are well out of the density (see attachment). This should be corrected, and in general I would advise double-checking the NCS restraints that are being used, as there are more differences between the three protomers than there should be for a C3 symmetric map and structure.
- There are two additional fairly strong densities near E225 and E227 – I am not sure whether it is worth commenting on these or not (perhaps not), but if I had to guess I would consider the possibility that ions might bind in this location.

Functional data

- The activity of C467R seems to be much lower than that of the wild type protein. It would be worth at least mentioning this in the text if it is indeed much less active (I may be overlooking something), as otherwise it is easy to miss, because the two panels of Fig.1 are shown with different scales on the Y axis.

Reviewer #3:

Remarks to the Author:

Synopsis and evaluation summary

The manuscript "A one-gate elevator mechanism for the human neutral amino acid transporter ASCT2" by Alisa A. Garaeva et al. presents open-inward structures of the human ASCT2 amino acid transporter. Building upon work published in 2018 by the same group it addresses the important question, how access to the substrate binding site is gated in this class of transporters. SLC1A transporters work by an elevator-like mechanism in which the so-called transport domain moves as a rigid body against a static scaffold domain. A matter of debate is whether access to the substrate binding site is controlled by the same gate loop on the extra- and intracellular side or whether separate gate loops exist on either side. This study presents a total of three structures obtained by single-particle cryo-EM at intermediate to low resolution that support the one-gate elevator hypothesis.

A single point mutation was introduced into ASCT2 to enable binding of the inhibitor TBOA that helped in obtaining inward-open conformations of the related EAAT transporters. Uptake experiments on proteoliposomes confirmed that the mutant protein changed substrate specificity and was indeed inhibited by this TBOA. The authors subsequently solved the structure of the mutant transporter both in presence and absence of inhibitor. Both structures show an inward-open conformation with loop HP2 acting as the gate on the intracellular side, supporting a one-gate elevator mechanism. However, somewhat surprisingly, neither the mutation, nor the inhibitor seemed to be critical in obtaining this conformation as indicated by a structure of ASCT2WT in the absence of inhibitor. Instead the authors propose that a lipid molecule could be involved in the gating process.

Despite the low resolution, the data appears solid and supports the authors' claims regarding the proposed one-gate elevator mechanism. However, the scope of the study is also limited by just focusing on the HP2 gate and providing little insight into functional aspects of this gating.

Again in favor of the authors is that they seem to be aware of the limits the low resolution sets and avoid to draw too many, too speculative conclusions.

Overall the manuscript is well written, though short, and the figures are of high quality. If the authors could at least expand the discussion of functional aspects in a revised manuscript, I would

consider publication in Nature Communications.

Major experimental issues

I have some concerns regarding the quality of the apo-ASCT2C467R structure. As visible in Sup. Fig. 2, the map shows clear anisotropy and I am concerned that the authors are overstating the resolution of the map. The FSC curves for the ASCT2C467R-TBOA map look significantly better. It is also a bit disappointing that the apo-ASCT2WT structure was not refined to a higher resolution. While the map and model seem to support the authors' claim that neither the C467R nor TBOA are essential to obtain the inward-open conformation, it is a bit surprising that the quality of this map is so much lower than that of apo-ASCT2C467R, especially since the number of particles is quite similar in both reconstructions.

In their analysis the authors only focus on the HP2 loop and I am wondering if there are other conformational differences that are not visible at 7 Å resolution. Furthermore, can the authors comment on this apparent preference for an inward-facing conformation in the absence of substrate? This topic has been briefly touched upon in their previous publication, but mainly in comparison to other transporter structures from the same family. Maybe it could be repeated in a short statement or discussed more from a functional point of view.

The role of lipid binding could also deserve a bit more attention to understand the actual gating mechanism. For example, have the authors tried to solve a structure in the presence of high concentrations of phospholipids and glutamine to see if lipids can enforce an open conformation in the presence of substrate.

Maybe these questions could also be addressed with molecular dynamics simulations.

Minor experimental issues

Can the authors comment on the linear accumulation of Asp into liposomes in the presence of TBOA?

Based on Sup. Fig. 5 the modeled position TBOA appears to be slightly off from the position in the map and potentially in the wrong orientation. How was the small molecule placed in the map? The occupancy also seems to be low since the signal appears to be quite weak.

Several other densities are mentioned, but obviously any interpretation would be very ambiguous at the given resolution.

While it is appreciated that structure validation reports were included, the structures and maps should have been deposited prior to submission of the manuscript and proper submission reports should have been provided.

Major editing issues

Fig. 2: Labels should be added for transmembrane helices to help identifying structural elements.

Sup. Fig. 1-3: If labels for structural elements presented in Sup. Fig. 4 should be added to be able to assess the fit between model and map at the given local resolution.

Sup. Fig. 5 should be moved to the main text.

TBOA should be spelled out once.

Minor editing issues (typos etc.)

Line 44: Whereas ... => While ...

Line 53: ... of a SLC1A transporter ... => ... of an SLC1A transporter ...

Line 128: ... potential inhibitors sites ... => ... potential inhibitor sites ...

Line 177: Confirm centrifugation speed. 20817 g appears somewhat low to pellet membranes.

Line 214: ... prior sample freezing ... => ... prior sample to freezing ...

Line 217: ... were applied on the grids ... => ... were applied onto the grids ...

Line 222: Prior data collection, best regions ... => Prior to data collection, the best regions ...

Line 235: ... and with a low-resolution estimations of the CTF fit ... => ... and with a low-resolution estimation of the CTF fit ...

Line 300: Trends Biochem Sci => Trends Biochem. Sci.

Chain A:

Chain B:

Chain C:

We thank the reviewers for the overall positive comments and the constructive suggestions and points raised. As requested we have expanded the overall format and lengthened the discussion. We hope we could address all issue to the reviewer's satisfaction. See our responses to every single point raised below.

Reviewer comments in italic
Answers to the reviewers' comments in blue.
Indicated lines refer to revised submission.

Reviewer #1

Garaeva, Guskov, Slotboom and Paulino present excellent results on the inward-open conformation of the neutral amino acid transporter ASCT2, which is of the same protein family as the glutamate transporters of the EAAT family. A good 3.6 Å resolution cryo-EM structure is obtained for a Cys367Arg mutation (mimicking the substrate site of EAAT's and now gaining Asp transport while loosing Gln transport) with negatively charged TBOA inhibitor and a lower resolution structure is obtained for the wt enzyme, showing the same inward-open state. The authors find the HP1 loop to gate also the inward oriented state, as it does for the outward-oriented states. They conclude that these transporters use the combination of an elevator mechanism and a single gate. This is very interesting from a mechanistic point of view for EAAT's specifically and transporters in general. The cryo-EM and functional studies in liposomes is beautifully done and provides an excellent basis for the report, and I have only manageable comments on the report:

Overall, the report is very short and could gain from being expanded a bit. For example to include a (much) closer comparison to the inward-open structure obtained by Hg crosslinking (Reyes et al., ref. 16).

Also the final remarks on drug ddiscovery using this mechanistic insight is very vague - please be more specific or end on a different note.

We have expanded the format of the report on several levels and included additional figures. The structure obtained by Reyes et al by Hg crosslinking represents an inward-facing occluded state of the protein. In our previous article (Garaeva et. al. 2018), where we determined the first ASCT2 inward-facing occluded state, we provided a comprehensive comparison with all available occluded/closed inward-facing states of SLC1A members. Here, we report the first inward-facing open state of a SLC1A member. For clarity we have mainly focused on comparing the open with the occluded inward-facing state of ASCT2. An overview on how further the transport domain is tilted in the open compared to other occluded inward-facing states can now be appreciated in Supplementary Figure 5a.

small comment:

Line 27: please explain "40% similarity" - this is an unclear term

We refer here to a 40% sequence similarity between EAAT2 and ASCT2, thus the resemblance of amino acids when between both structures. To avoid the misunderstanding, we have rewritten the sentence as follow (lines 35-36):

"While EAATs and ASCTs share significant similarities in sequence (~40%) and structure, they are mechanistically different: ..."

Reviewer #2

Summary:

Garaeva et al. present three structures of the Alanine Serine Cysteine transporter 2 (ASCT2) in the inward open state – two of a point mutant, C467R, in the presence and absence of ligand, and one of the wild-type protein in the apo state. The best of these, the C467-TBOA structure, is resolved to 3.6Å. All three structures are very similar in conformation, and differ from the inward occluded structure of the same protein solved by Dr. Paulino's lab primarily in the movement of a loop, HP2, which occludes the ligand binding site in the previous structure, while exposing it in all three structures presented in this manuscript. This is significant, because this is the same loop that also acts as a gate to entry of the ligand in the outward-open structures of other members of the SLC1A family, demonstrating that SLC1A transporters operate via a one-gate elevator mechanism, rather than a two-gate mechanism as previously theorized. In addition to the structural data, the authors also present functional assays demonstrating that a single point mutation, C467R, is sufficient to switch specificity of the transporter from neutral amino acids to acidic substrates.

Main Impressions:

This is an excellent and thorough study, with high quality structural data addressing a clearly posed question – how do SLC1A transporters release substrates into the cell after transport across the membrane? Having previously solved the structure of an inward occluded form of the same transporter (wt-ASCT2 bound to glutamine), the authors were in an excellent position to resolve this question. The authors first attempted to use a bulky substrate (TBOA) to preclude the inner gate from closing, trapping the transporter in the inward open form. This approach had been previously used to capture the outward-open state of another SLC1A transporter, EAAT1. However, this substrate did not appreciably inhibit transport of glutamine by ASCT2, which the authors attributed to the specificity of ASCT2 for neutral rather than acidic substrates. To address this, Garaeva et. al. introduced a point mutation in the ligand binding pocket (C467R) intended to convert ASCT2 to a transporter for acidic rather than neutral amino acids, and also make the transporter amenable to inhibition by TBOA. This approach was successful (the mutant receptor now transports aspartate preferentially to glutamine and is inhibited by TBOA), although it appears to be much less active overall than the wild type protein. The authors go on to solve the structure of ASCT2-C467R in the presence of TBOA at 3.6Å, and the density map and model are overall of excellent quality (see minor comments below). The density for the ligand however is very poor, and it is not clear to me based on the density that it is bound at the assigned location with any appreciable occupancy. However, ASCT2 is clearly in the inward-open state,

based on comparison of the map and model with the inward occluded state of the protein – the movement of the HP2 loop to uncover the ligand binding cavity is very clear. In order to test whether the added ligand is indeed having the expected effect, and whether the introduced mutation is affecting the conformation of HP2, the authors solved two additional, lower resolution structures, of C467R in the absence of ligand, and of the wild-type protein in the apo state. Both of these structures look identical to the notionally ligand-bound mutant structure. Given the absence of ordered density for the TBOA molecule, this suggests that the ligand was perhaps not bound in the initial dataset, or at least that there is little evidence for TBOA binding in the current study. This is not of critical importance however, as the protein is clearly in the inward-open configuration in all three states, and in all three states opening is mediated by movement of HP2, which answers the question that the authors set out to address.

I do feel the apparently unexpected finding that the inner gate is open even in the absence of added inhibitor could do with a little further discussion.

We have slightly expanded the discussion.

See lines 129-132: “Thus, neither the presence of the inhibitor nor the point mutation itself is responsible for gate opening on the intracellular side. It appears that an inward-open state of ASCT2 is favoured in the given conditions.”

*Later we also hint that the co-purified lipid near the hairpins might act as an inhibitor and arrest the protein in the inward-open state. See lines 193-197: “In particular the well-defined site for a co-purified lipid from *Pichia pastoris* found between the hairpins HP1 and HP2 (Fig. 3b) could be a potential target location for the development of binders that could block ASCT2 activity and prevent glutamine supply in cancer cells. Such lipid-like binders could act as inhibitor, arresting the protein in the observed inward-open state of ASCT2.”*

Overall, while I have a few minor technical comments below that I would like to see addressed, this is an excellent study that I am happy to recommend for publication.

Specific comments:

Image processing

Overall, the image processing workflow is very clear and easy to follow – Supp Figs1-4 could serve as a guide for others in the field in how to present this information in an easily digestible fashion! I particularly appreciate that the mask used in FSC calculation is depicted on the FSC curve, and that the per-TM density figures have been calculated in a manner that does not recontour the density to close over holes in the mesh (which is quite common when using Pymol). I have a couple of minor queries:

- It is stated that the map was obtained using C3 symmetry, but the supplied map seems to have some differences between the three protomers – mostly in the distribution of noise peaks. This should not be the case if symmetry was enforced. Perhaps some procedure after refinement (e.g. local resolution filtering) introduced these features?*

A C3 symmetry was imposed during reconstruction and we agree with the reviewer that one would expect a perfect symmetrical rendered map and are intrigued. We did not apply any local resolution filtering. Since the differences are very minimal and mostly observed for noise around the protein we suspect that it might derive from a non-perfect C3-symmetric mask used during post-processing.

• In the text, the global resolution is listed as 3.61 Å, but in Table 1 the resolution range for the map is listed as 4.2-3.4. To be clear, the map has been filtered to 3.6Å, correct? If this is meant to be the local resolution range, I would suggest specifying it as such (as opposed to being the resolution range of spatial frequencies included in the final map).

The reviewer is right, we meant here the local resolution range as depicted from the map colored with respect to its local resolution. The final deposited map has been low-pass filtered to 3.6Å. This have indicated it in the table as suggested.

• CTF refinement in Relion is mentioned, but further details should be supplied for completeness – was this just per particle defocus refinement? Beam tilt refinement?

Both were applied, per particle CTF refinement and beam tilt refinement. This is now better specified in material and methods (see lines 318-320).

Model building and ligand assignment

• There is very little support in the density map for the assignment of the TBOA ligand – there is a small peak where the ligand has been assigned, but this peak is smaller than many of the noise peaks in the micellar region, and much weaker than any of the unassigned lipid/cholesterol/detergent densities. I think is worth mentioning the presence of this peak, but without stronger evidence (e.g. a difference density with an apo map of comparable resolution), I don't think it is justifiable to include the ligand in the deposited model. In addition, the ligand appears to be assigned as covalently bound to R467, with a distance of 1.3Å from O4 of TBOA to NE of R467. Is this intended/expected?

We agree with the reviewer that the density is very weak and we have thus not included TBOA in the deposited model. We discuss that it probably derives from a low occupancy. See lines 117-120: "In the substrate-binding site we observe a patch of non-protein density close to R467.

While the density is too weak to allow for an unambiguous assignment, it most likely represents a TBOA molecule, although the occupancy of the site appears low (Supplementary Fig. 5b)."

Interestingly, the density of TBOA solved in X-ray structures of ASCT2 homologues EAAT1 and GltPh in their outward-open states is comparably weak (see ref 15,19). In line with the maps of apo-ASCT2 (wildtype and C467R mutant), we conclude that TBOA is not required to arrest the protein in the open state (see lines 129-132). To guide the reader that the additional density potentially represents TBOA we keep it in supplementary figure 5b.

The presumably covalent bound between TBOA and R467 is not present. We apologize for this very unfortunate mistake, as we have provided the reviewer with a wrong, intermediate, pdb model. We have attached the correct version were the reviewer will be able to appreciate that TBOA is better fitted (similar to as observed for its homologues in the outward-open states) and further away from R467.

• There are several cis-peptide bonds of non-Pro residues which should be fixed – check residues 122, 124, 428 & 367. 428 in particular needs attention, as it is located in HP2, near the ligand binding site.

This issue is caused by the same mistake reported above (you had received the wrong not final refined model). In the final version all non-pro cis-peptides are fixed.

• The conformation of HP2 differs between the three protomers – it is more or less concordant with the density in chain A (except for the cis peptide at A428 mentioned above), but in chains B and C P432 and I431 are well out of the density (see attachment). This should be corrected, and in general I would advise double-checking the NCS restraints that are being used, as there are more differences between the three protomers than there should be for a C3 symmetric map and structure.

This issue is caused by the same mistake reported above (you had received the wrong not final refined model). In the final version the model fits equally well the density in all three protomers. We apologize once again for the unnecessary confusion.

• There are two additional fairly strong densities near E225 and E227 – I am not sure whether it is worth commenting on these or not (perhaps not), but if I had to guess I would consider the possibility that ions might bind in this location.

We highly appreciate the thorough revision of the data by the reviewer and agree that there is a strong peak putatively coordinated by S194, N195, E225 and F201' and E227' of the adjacent protomer near the antennae. However, since the coordination distances are around 4 Å (see attached image) we are puzzled by what it could be as it does not fit any typical ion coordination. We thus avoid to over interpret, but show the density in an additional panel in Supplementary Figure 7g, and have included its description in the manuscript. See lines 160-165: “Lastly, a density was found on the extracellular side of the scaffold domain. It is located between the N- and C-termini of the ASCT2 antennae (TM4b,c-loop) of two adjacent protomers (Supplementary Fig. 7a,g), with distances to neighbouring side chains of ~4Å. The shape of the density suggests that it might represent a hydrated bound ion, but it is not possible to unambiguously determine the identity from the density map.”

Functional data

- *The activity of C467R seems to be much lower than that of the wild type protein. It would be worth at least mentioning this in the text if it is indeed much less active (I may be overlooking something), as otherwise it is easy to miss, because the two panels of Fig.1 are shown with different scales on the Y axis.*

We believe that it is not possible to simply state that the mutant is much less active than the wild-type, because we would be comparing glutamine by the wildtype with aspartate uptake by the mutant. ASCT2-C467R doesn't support glutamine transport (so indeed is less active in glutamine transport than the wildtype), but ASCT2-wt doesn't transport aspartate (so the mutant is more active than the wildtype in transport of this substrate). If the WT/glutamine uptake is compared to the mutant/aspartate uptake, indeed the latter is lower. We re-wrote the paragraph, see lines 90-95: "Although the uptake rate of aspartate by ASCT2_{C467R} was slower than that of glutamine by ASCT2_{wt}, the results confirm that a single point mutation in the substrate binding site was sufficient to switch the substrate specificity of ASCT2 from neutral to acidic amino acids. In addition, aspartate transport could be suppressed by TBOA (Fig. 1b), which opened the possibility to use the compound to trap the transporter in an open state."

Reviewer #3:

Synopsis and evaluation summary

The manuscript "A one-gate elevator mechanism for the human neutral amino acid transporter ASCT2" by Alisa A. Garaeva et al. presents open-inward structures of the human ASCT2 amino acid transporter. Building upon work published in 2018 by the same group it addresses the important question, how access to the substrate binding site is gated in this class of transporters. SLC1A transporters work by an elevator-like mechanism in which the so-called transport domain moves as a rigid body against a static scaffold domain. A matter of debate is whether access to the substrate binding site is controlled by the same gate loop on the extra- and intracellular side or whether separate gate loops exist on either side. This study presents a total of three structures obtained by single-particle cryo-EM at intermediate to low resolution that support the one-gate elevator hypothesis.

A single point mutation was introduced into ASCT2 to enable binding of the inhibitor TBOA that helped in obtaining inward-open conformations of the related EAAT transporters. Uptake experiments on proteoliposomes confirmed that the mutant protein changed substrate specificity and was indeed inhibited by this TBOA. The authors subsequently solved the structure of the mutant transporter both in presence and absence of inhibitor. Both structures show an inward-open conformation with loop HP2 acting as the gate on the intracellular side, supporting a one-gate elevator mechanism. However, somewhat surprisingly, neither the mutation, nor the inhibitor seemed to be critical in obtaining this conformation as indicated by a structure of ASCT2WT in the absence of inhibitor. Instead the authors propose that a lipid molecule could be involved in the gating process.

Despite the low resolution, the data appears solid and supports the authors' claims regarding the proposed one-gate elevator mechanism. However, the scope of the study is also limited by just focusing on the HP2 gate and providing little insight into functional aspects of this gating.

Again in favor of the authors is that they seem to be aware of the limits the low resolution sets and avoid to draw too many, too speculative conclusions.

Overall the manuscript is well written, though short, and the figures are of high quality. If the

authors could at least expand the discussion of functional aspects in a revised manuscript, I would consider publication in Nature Communications.

We thank the reviewer for the positive assessment of our work. As suggested we have expanded the overall format of the manuscript.

Major experimental issues

I have some concerns regarding the quality of the apo-ASCT2C467R structure. As visible in Sup. Fig. 2, the map shows clear anisotropy and I am concerned that the authors are overstating the resolution of the map. The FSC curves for the ASCT2C467R-TBOA map look significantly better.

We agree with the reviewer that there is some anisotropy present in the data obtained for apo-ASCT2-C467R. However, we disagree that it would lead to a severe overestimation of the resolution claimed. As depicted by the 3D FSC plot shown in supplementary figure 2g, all 4 lines (FSC calculated for global and X, Y, Z directions) - while oscillating - overlap over the majority of the resolution range and largely coincide at the FSC threshold of 0.143. Moreover, we have used the ASCT2-C467R map at 3.6Å for model building and used the apo-ASCT2-C467R only for comparisons. I hope that the reviewer agrees that the interpretation we draw from such a map comparison is not jeopardized by the slight anisotropy nor the indicated resolution range.

It is also a bit disappointing that the apo-ASCT2WT structure was not refined to a higher resolution. While the map and model seem to support the authors' claim that neither the C467R nor TBOA are essential to obtain the inward-open conformation, it is a bit surprising that the quality of this map is so much lower than that of apo-ASCT2C467R, especially since the number of particles is quite similar in both reconstructions.

We share the same disappointment with the reviewer. ASCT2-wt appeared less stable without its substrate and despite several attempts we always faced difficulties in obtaining good cryo-EM data for it. Our interpretation is that, despite the similar number of particles, the quality of particles is worse for ASCT2-wt than for the mutant, leading to a lower resolution map.

In their analysis the authors only focus on the HP2 loop and I am wondering if there are other conformational differences that are not visible at 7 Å resolution. Furthermore, can the authors comment on this apparent preference for an inward-facing conformation in the absence of substrate? This topic has been briefly touched upon in their previous publication, but mainly in comparison to other transporter structures from the same family. Maybe it could be repeated in a short statement or discussed more from a functional point of view.

We have carefully compared other regions between the ASCT2-wt apo map and the ones obtained for the mutant and could not find any significant differences. However, as pointed out by the reviewer, while 7 Å resolution allows to compare the relative position of secondary structure elements as helices, hairpins and loops, the resolution is not sufficient to draw any conclusion on the detailed position of amino acids and the orientation of their sidechains.

Analogously as done for the ASCT2-C467R-TBOA dataset (see Supplementary figure 1d), we carefully checked for conformational heterogeneity, in which a single protomer within the trimer might adopt a different conformation. One approach was to refine the last set of apo-ASCT2-wt particles without imposing a symmetry, which however yielded a too low-resolution map that was

not interpretable. In our second approach we performed 3D classification on the individual protomers of a single transporter obtained by signal subtraction and symmetry expansion. Via this approach each protomer is treated individually, tripling the number of particles. The 3D classification followed by refinement rendered only one interpretable class. The protomers were all in the same conformation, namely the inward-open state as observed when refined as a trimer (see new workflow in Supplementary figure 3d).

As for our previous work, where in presence of its substrate ASCT2 was only found in an inward-occluded state, we were surprised to observe that in absence of its substrate ASCT2 (wt and mutants) appears to again only adopt one conformation, here the inward-open state. We assume that under the given conditions (buffer, detergent, freezing conditions and time) the inward state may be the lowest energy state of the protein. See lines 103-105: “Notably, only a single conformational state was captured, in which all protomers adopted the same conformation, indicating a preferred state of the transporter in the given conditions (Supplementary Figs. 1d).” We also gently speculate that the co-purified lipid found near the hairpins might act as an inhibitor and arrest the protein in the observed inward-open state. See lines 193-200: “In particular the well-defined site for a co-purified lipid from *Pichia pastoris* found between the hairpins HP1 and HP2 (Fig. 3b) could be a potential target location for the development of binders that could block ASCT2 activity and prevent glutamine supply in cancer cells. Such lipid-like binders could act as inhibitor, arresting the protein in the observed inward-open state of ASCT2.”

The role of lipid binding could also deserve a bit more attention to understand the actual gating mechanism. For example, have the authors tried to solve a structure in the presence of high concentrations of phospholipids and glutamine to see if lipids can enforce an open conformation in the presence of substrate.

Maybe these questions could also be addressed with molecular dynamics simulations.

We have extended the discussion on the lipids (see lines 182-200). We thank the reviewer for the suggestion of trying to solve a structure in presence of high phospholipid concentration and glutamine. However, we hope the reviewer agrees that in the interest of time it will be impossible to include it in this study.

Minor experimental issues

Can the authors comment on the linear accumulation of Asp into liposomes in the presence of TBOA?

Although the observed activity curve of ASCT2-C467R with aspartate and TBOA appears linear, it is not, the residual uptake in the presence of the inhibitor is simply very slow.

Based on Sup. Fig. 5 the modeled position TBOA appears to be slightly off from the position in the map and potentially in the wrong orientation. How was the small molecule placed in the map? The occupancy also seems to be low since the signal appears to be quite weak.

The TBOA molecule was primarily fitted manually and guided by the position found with other homologs (ref 15,19). The aspartate-derivate is facing the amino-acid binding site while the benzyloxy group is facing to the intracellular side. We agree with the reviewer that the density is

weak and might indicate a low occupancy, which goes in line with the fact that TBOA binding is not required to capture the inward-open state. We have now omitted the TBOA in the deposited model (see also response to reviewer 1).

Several other densities are mentioned, but obviously any interpretation would be very ambiguous at the given resolution.

While it is appreciated that structure validation reports were included, the structures and maps should have been deposited prior to submission of the manuscript and proper submission reports should have been provided.

We apologize that the reviewer had no access to the maps and models, which we are always happy to immediately share to provide full transparency. In fact, we did provide all the information, but it seems to have only been shared with reviewer #2. As a side note: The deposition of cryo-EM maps and models is not mandatory yet. As the refinement process of cryo-EM data is rather time-consuming and not as static as for X-ray crystallography, valuable feedback is often given during the review process, and new software may be released that might help to improve the image processing, we prefer to only deposit the final data at the end of the revision trajectory. The maps and models have been deposited and the respective IDs were added to the manuscript.

Major editing issues

Fig. 2: Labels should be added for transmembrane helices to help identifying structural elements. Sup. Fig. 1-3: If labels for structural elements presented in Sup. Fig. 4 should be added to be able to assess the fit between model and map at the given local resolution.

To guide the reader better and label all transmembrane helices we have created additional panels and added them to Figure 2.

Sup. Fig. 5 should be moved to the main text.

We have integrated former Supplementary Figure 5 a-b into Figure 4. We have however kept the weak TBOA density in the supplementary figure, as we have not deposited TBOA with the model.

TBOA should be spelled out once.

It is now included in material and methods (see line 281-282).

Minor editing issues (typos etc.)

Line 44: Whereas ... => While ...

Corrected

Line 53: ... of a SLC1A transporter ... => ... of an SLC1A transporter ...

Corrected

Line 128: ... potential inhibitors sites ... => ... potential inhibitor sites ...

Corrected

Line 177: Confirm centrifugation speed. 20817 g appears somewhat low to pellet membranes.
It is the centrifugation speed used.

Line 214: ... prior sample freezing ... => ... prior sample to freezing ...
We corrected is as “ prior to sample freezing”

Line 217: ... were applied on the grids ... => ... were applied onto the grids ...
Corrected

Line 222: Prior data collection, best regions ... => Prior to data collection, the best regions ...
Corrected

Line 235: ... and with a low-resolution estimations of the CTF fit ... => ... and with a low-resolution estimation of the CTF fit ...
Corrected

Line 300: Trends Biochem Sci => Trends Biochem. Sci.
Corrected

Reviewers' Comments:

Reviewer #2:

Remarks to the Author:

I thank the authors for their response to my review. The authors have addressed all my comments in satisfactory detail, and I am happy to recommend the manuscript for publication in Nature Communications as is. I would only add that it may be worth adding a sentence or two to comment on the preprint that has been deposited by Pfizer (<https://www.biorxiv.org/content/10.1101/622563v1>) during the course of review, which describes the outward facing conformation of ASCT2, as this would seem complementary to the results in your manuscript.

Reviewer #3:

Remarks to the Author:

Synopsis and evaluation summary

Garaeva et al. present a revised version of their manuscript "A one-gate elevator mechanism for the human neutral amino acid transporter ASCT2" submitted to Nature Communications. The authors have extended the text of the manuscript and added additional figures to the main text and panels in the Supplementary Figures. The models have been further refined as well and MolProbity validation statistics have slightly improved. Notably, TBOA has been removed from the final deposited model determined in the presence of TBOA due to the weak density observed in the map. As with the densities for potentially bound lipids, the conclusions have been toned down and the authors are careful not to overinterpret them, while still acknowledging their presence and hinting at possible implications. With this, the major critique of me, as well as the other reviewers, has been sufficiently addressed. It would have been nice if other questions had been addressed, like better resolution for apo wild-type ASCT2, and the role of lipids, but those are not critical for this manuscript and I agree that these additional experiments would probably require too much time. I therefore support publication in the current form and look forward to hopefully seeing these questions addressed in later publications. Some remaining issues in editing as listed below can be addressed during proof reading.

Major editing issues

The sentence starting in line 103 (... in which all protomers adopted the same conformation, ...) should be re-written to clearly indicate that the authors confirmed this by refinement without symmetry imposed and the statement holds true despite applying C3 symmetry for the final map refinement.

The sentence starting in line 131 should be modified to "... favoured in the given conditions without L-glutamine added." to make it immediately clear to the reader what is different compared to the preparation of the sample that yielded the inward-occluded structure. Maybe also include this in the methods section.

Include a statement in the discussion that the lower resolution of the apo wild-type map can be attributed to a more heterogeneous sample or lower particle quality and that the C467R has some stabilizing effect.

Minor editing issues (typos etc.)

Line 164: ... identity form the density ... => ... identity from the density ...

Line 235: ... 10 ml of BMGY medium was inoculated ... => ... were inoculated ...

Line 240: ... resuspended in 500µl lysis buffer ... => ... 500 µl ...

Line 249: Large scale ... => Large-scale ...

Line 249: Model building was carried in Coot ... => Model building was carried out in Coot ...

Line 442: Journal name for reference 34 is missing

Line 516: ... x, y, and z axis displayed ... => ... x, y, and z axis are displayed ...

Line 530: ... x, y, and z axis displayed ... => ... x, y, and z axis are displayed ...

Line 540: ... x, y, and z axis displayed ... => ... x, y, and z axis are displayed ...

We thank the reviewers for the overall positive comments and the constructive suggestions and points raised. We corrected all remarks. See our responses to every single point raised below.

Reviewer comments in italic

Answers to the reviewers' comments in blue.

Indicated lines refer to revised submission.

Reviewer #2

I thank the authors for their response to my review. The authors have addressed all my comments in satisfactory detail, and I am happy to recommend the manuscript for publication in Nature Communications as is. I would only dd that it may be worth adding a sentence or two to comment on the preprint that has been deposited by Pfizer (<https://www.biorxiv.org/content/10.1101/622563v1>) during the course of review, which describes the outward facing conformation of ASCT2, as this would seem complementary to the results in your manuscript.

In fact, we had it referenced in the last revised version obtained by the reviewer. However, the editorial decision from Nature Communication is to not reference this preprint anymore as it has not been peer-reviewed nor the coordinated are out to make a suitable comparison.

Reviewer #3

Synopsis and evaluation summary

Garaeva et al. present a revised version of their manuscript "A one-gate elevator mechanism for the human neutral amino acid transporter ASCT2" submitted to Nature Communications. The authors have extended the text of the manuscript and added additional figures to the main text and panels in the Supplementary Figures. The models have been further refined as well and MolProbity validation statistics have slightly improved. Notably, TBOA has been removed from the final deposited model determined in the presence of TBOA due to the weak density observed in the map. As with the densities for potentially bound lipids, the conclusions have been toned down and the authors are careful not to overinterpret them, while still acknowledging their presence and hinting at possible implications. With this, the major critique of me, as well as the other reviewers, has been sufficiently addressed. It would have been nice if other questions had been addressed, like better resolution for apo wild-type ASCT2, and the role of lipids, but those are not critical for this manuscript and I agree that these additional experiments would probably require too much time. I therefore support publication in the current form and look forward to hopefully seeing these questions addressed in later publications. Some remaining issues in editing as listed below can be addressed during proof reading.

Major editing issues

The sentence starting in line 103 (... , in which all protomers adopted the same conformation, ...) should be re-written to clearly indicate that the authors confirmed this by refinement without symmetry imposed and the statement holds true despite applying C3 symmetry for the final map refinement.

Corrected, see lines 103-107 of revised manuscript.

The sentence starting in line 131 should be modified to "... favoured in the given conditions without L-glutamine added." to make it immediately clear to the reader what is different compared to the preparation of the sample that yielded the inward-occluded structure. Maybe also include this in the methods section.

Corrected, see lines 132-134.

Include a statement in the discussion that the lower resolution of the apo wild-type map can be attributed to a more heterogeneous sample or lower particle quality and that the C467R has some stabilizing effect.

This is discussed in more detail in the respective result section (see lines 128-130).

Minor editing issues (typos etc.)

Line 164: ... identity form the density ... => ... identity from the density ...

Corrected

Line 235: ... 10 ml of BMGY medium was inoculated ... => ... were inoculated ...

Corrected

Line 240: ... resuspended in 500µl lysis buffer ... => ... 500 µl ...

Corrected

Line 249: Large scale ... => Large-scale ...

Corrected

Line 249: Model building was carried in Coot ... => Model building was carried out in Coot ...

Corrected

Line 442: Journal name for reference 34 is missing

Corrected

Line 516: ... x, y, and z axis displayed ... => ... x, y, and z axis are displayed ...

Corrected

Line 530: ... x, y, and z axis displayed ... => ... x, y, and z axis are displayed ...

Corrected

Line 540: ... x, y, and z axis displayed ... => ... x, y, and z axis are displayed ...

Corrected